# Machine Learning-Based Regression Models for Ironmaking Blast Furnace Automation

**Ricardo A. Calix [1], Orlando Ugarte [2], Tyamo Okosun [2] and Hong Wang [3],***

[1] Department of Computer Information Technology, Purdue University Northwest, Hammond, IN 46323, USA; rcalix@pnw.edu

[2] Center for Innovation through Visualization and Simulation (CIVS) and Steel Manufacturing Simulation and Visualization Consortium (SMSVC), Purdue University Northwest, Hammond, IN 46323, USA; ougarte@pnw.edu (O.U.); tokosun@pnw.edu (T.O.)

[3] Buildings and Transportation Science Division, Oak Ridge National Laboratory, Knoxville, TN 37932, USA

\* Correspondence: wangh6@ornl.gov; Tel.: +1-(219)-989-3157

**Abstract:** Computational fluid dynamics (CFD)-based simulation has been the traditional way to model complex industrial systems and processes. One very large and complex industrial system that has benefited from CFD-based simulations is the steel blast furnace system. The problem with the CFD-based simulation approach is that it tends to be very slow for generating data. The CFD-only approach may not be fast enough for use in real-time decisionmaking. To address this issue, in this work, the authors propose the use of machine learning techniques to train and test models based on data generated via CFD simulation. Regression models based on neural networks are compared with tree-boosting models. In particular, several areas (tuyere, raceway, and shaft) of the blast furnace are modeled using these approaches. The results of the model training and testing are presented and discussed. The obtained $R^2$ metrics are, in general, very high. The results appear promising and may help to improve the efficiency of operator and process engineer decisionmaking when running a blast furnace.

**Keywords:** XGBoost; computational fluid dynamics; steel blast furnace; machine learning; regression

## 1. Introduction

The ironmaking blast furnace is a countercurrent chemical reactor that is designed to drive high-temperature CO- and $H_2$-reducing gases through a packed bed of burden material. The burden is composed of iron oxides and coke (carbon) fuel. The hot reducing gas, which is generated via the combustion of fuels and oxygen-enriched hot blast air delivered through nozzles called tuyeres in the lower part of the furnace, serves to reduce the oxides into metallic iron before melting it into liquid. This liquid pig iron collects in the hearth and can then be cast and transferred to other stages of the steelmaking process. This process is responsible for most new iron production in North America, accounting for 72% of ironmaking in 2020 and a significant majority of steelmaking worldwide [1]. A wide range of operating parameters determine the performance of the blast furnace process, which presents challenges to operators in maintaining stability and reducing coke consumption (which is generally the most expensive fuel and is responsible for a large majority of the furnace's $CO_2$ emissions).

Key to this effort is maintaining process stability by selecting the proper operating conditions. One of the most critical guiding metrics in this regard is referred to as the operating window, often defined by the minimum safe temperatures of the furnace top gas and the temperature of combustion gases in the furnace raceway. The top gas temperature must be maintained above 100 °C to avoid the condensation of water vapor and the raceway flame temperatures must be high enough to supply thermal energy for both

reduction reactions and melting (generally over 1800 °C for North American operations). The operating parameters influencing these two variables are inversely correlated, and those that increase flame temperature generally decrease top gas temperature and vice versa. If either of these temperatures are too low, instability in the process can result, which has significant associated cost and operational risks. As a result, understanding the expected position of the furnace within this window for a given set of conditions in advance is critical for operators. In addition to raceway flame temperature and top gas temperature, which govern the operating window, several other factors can govern decisionmaking for blast furnace operators. Tuyere exit velocity can determine the size of the raceway cavity and the depth of gas penetration. A furnace pressure drop (from blast pressure to top pressure) can determine how much blast air can be supplied, therefore influencing the production rate. Injected pulverized coal burnout can influence the coke consumption rate of the furnace and correspondingly the cost of operating the process. From these parameters, which can be directly predicted by modeling techniques, other operating conditions can often be inferred, allowing for informed decisionmaking using blast furnace operators.

With these parameters in mind, and understanding that even small improvements to coke consumption rates can significantly affect operating costs and $CO_2$ emissions, operators have turned to modeling methods for predicting the performances of their furnaces under various operating conditions. These conditions include industrial rules of thumb that have been established over decades of experience and experimentation, heat and mass balance models designed to provide theoretical estimations [2], and the most recent addition to these tools, computational fluid dynamics (CFD) simulation. Simulation and modeling approaches provide faster and more flexible results than experimental testing, but CFD can also account for additional physical detail compared with conservation calculation approaches, including handling competing reaction rates, combustion kinetics, and the effects of fluid flow patterns.

Unfortunately, CFD-based simulation approaches generally require significant computing time for each new solution generated, with results and data being generated too slowly for use in real-time or on-the-fly decisionmaking. To address this issue, in this work, the authors propose the use of machine learning techniques to train and test reduced-order models based on data generated via CFD simulation. Machine learning (ML) has already been used in large-scale industrial applications with some success. Some studies of note include [3–9]. Specifically, for the steelmaking industry, ML has been used in several ways to develop predictive models. Many models use ML techniques to build regression models for blast furnace output parameter calculation. Other approaches have used deep learning techniques for the image classification of furnace input material detection and analysis. An additional discussion of ML as applied to large industrial applications can be seen in Section 2. In our work, we propose the use of CFD-based, data-driven ML models for use in the prediction of blast furnace output parameters. In particular, several areas of the blast furnace, each corresponding to a specific phenomenological region of the process, are separately modeled using machine learning-based approaches. A CFD model of each of these areas predicted phenomena under different operating conditions, with the results being subsequently used to train machine learning-based regression models. Regression models based on neural networks are compared with tree-boosting models, and the results of the model training and testing are presented and discussed. The obtained $R^2$ metrics are, in general, very high. The results appear promising and may help to improve the efficiency in decisionmaking when running a steel blast furnace. Two approaches were used for the modeling. One approach used neural network-based architectures, and the other one used XGBoost. The results looked promising for both approaches. In the end, both models provided advantages and disadvantages. The XGBoost models are more accurate for within-range data than the neural network models. However, XGBoost techniques do not perform well in attempts to extrapolate outside of the range values. For the case of extrapolation, neural networks seem to perform much better. Extrapolation to outside

of the range output prediction is desirable for this study, and as such, both approaches were used.

$R^2$ metrics were used to measure the performance of the regression models. Additionally, correlation coefficient matrices were also used to visualize the importance of the features in predicting the output values.

## 2. Literature Review

### 2.1. Machine Learning for Large Scale Industrial Applications

Machine Learning (ML) has been used in large-scale industrial applications with some success. Some studies of note include [3–9]. In [3], the authors provide a survey of how data science can and should be applied to the steelmaking industry and to blast furnaces in particular. The paper discusses several ML-based models that have been used for blast furnace predictive modeling. The paper is very general and examines many issues that range from data management to modeling. Abhale et al. [4] published another survey-type paper discussing many modeling approaches for blast furnace modeling, focusing on the historical evolution of blast furnace modeling. It provides a classification of different modeling approaches from physics that are based on data-driven methods. It classifies ML models into the category of "Data Driven Models".

Beyond survey-type papers, there have also been some more specific modeling studies in the application of machine learning for steelmaking processes. In [5], for instance, the authors proposed an ML-based performance prediction model of a basic oxygen steel reactor based on the operating parameters. In their work, the authors performed a correlation analysis to study the relationship between the inputs and outputs used to estimate the performance of the reactor. They also used neural networks to train a regression model to predict decarburization rates. The reported results appear promising. In [6], the authors performed a study on the analysis of particle size distribution of coke on the blast furnace belt using object detection. In particular, they used a pre-trained deep learning-based image object detection framework called YOLO. They reported that the object detection and size analysis are effective when using this pre-trained deep learning-based model.

In [7], the authors used regression modeling to predict blast furnace output parameters. In particular, the authors proposed the predictive modeling of blast furnace gas utilization rates using different methods. The authors found that support vector regression (SVR) models perform well for this task. Another similar study is [8]. Here, the authors performed a comparison of data-driven prediction models for a comprehensive coke ratio of the blast furnace. The authors found that the best methods to predict the coke ratio are SVR and multiple linear regression methods. They also used AdaBoost as well in their study and found that it could be as effective as the other models.

In [9], the authors performed a more specific literature review to train models for the prediction of hot metal silicon content in blast furnaces. The authors discuss several ML algorithms such as support vector machines (SVMs) and neural networks. They discuss common inputs and outputs for these types of models as well as the challenges and future directions.

Studies that propose replacing CFD simulation approaches with machine learning also include [10]. In their work, the authors proposed a model to predict airfoils using deep learning. Part of the data used to train their neural network is generated using a solver called OpenFOAM, which is an open-source CFD software. The authors concluded that their approach was efficient and accurate.

One very interesting study was performed by [11]. In their work, the authors proposed new ideas on how to integrate machine learning into CFD simulation software. The authors let neural networks determine algorithms that can be integrated into CFD software. The study involved the replacing of components of traditional solvers with data-driven interpolation, which allows for the acceleration of highly computationally intense solvers such as direct numerical simulation (DNS), which are generally greatly impacted by the computational mesh resolution. The approach was tested in computationally sped-up

studies against the DNS modeling of two-dimensional turbulent flows with promising results. Another similar approach is discussed in [12]. In this work, the authors also used machine learning to improve CFD-based simulations. The goal of their study was to use ML first to select the initial conditions for the CFD software. In this way, the CFD software can converge to an accurate solution faster without affecting the performance.

*2.2. Regression Modeling*

In statistics and machine learning, the question of how well a model performs for the data is sometimes described in terms of the bias and variance trade-off [13]. In this context, bias means that the model cannot learn the true relationship in the data. For example, a linear regression cannot fit a curve; thus, it has bias. Neural network-based models, for instance, can fit the training data so well that they can prove unable to adapt upon encountering new unseen data. This is called overfitting. This difference in fitting between the train set and the test set is sometimes called the variance in the context of overfitting. Thus, an overfitted model on the train set may have a higher variance because it does not generalize well to the unseen test set. Therefore, the goal when building a regression model for your data is to have both good bias and variance in the context of fitting the data.

There are several approaches to fit models to regression data. Three important methods are linear regression, regression trees, and neural networks. The next sections discuss neural network-based regression and regression trees leading up to XGBoost.

2.2.1. Regression Trees

As previously indicated, regression trees [14] can fit nonlinear data. Regression trees are a type of decision tree in which the predicted values are not discrete classes but are instead real valued numbers. The key concept here is to use the features from the samples to build a tree. The nodes in the tree are decision points that lead to leaves. The leaves contain values that are used to determine the final predicted regression value.

For example, if a selected feature ranges from 0 to 255, a threshold value, such as 115, can be selected to decide which child node of the parent node should be moved to. In the simplest case, each parent node in a regression tree has two child nodes. Once the selected feature's threshold is established, this cutoff is used to help select the output regression value. In the simplest case, the child node on the left can accumulate all of the output values that correspond to the values below the threshold for the given feature. Concurrently, the right node collects those above the threshold. Simply averaging values in each node can provide an initial predicted value. However, this method is not the best solution. The goal in training regression trees is to try multiple thresholds and select the ones that provide the best results. Through iterative optimization and comparison of predicted values to real values, optimal regression trees can be obtained.

Regression trees start from the top and work their way down to the leaf nodes. Leaf nodes are used to select output values. Nonleaf nodes are typically decision nodes that are based on the optimal threshold for the range of values in that feature. The key question is how to select which optimal features to use and how to determine the optimal threshold value for a given node.

How are features and thresholds selected? One simple way is to iteratively try different thresholds and then try predicting an output value with each regression tree model candidate. For each predicted output for a given tree candidate, the approach compares the predicted value with the real value and selects the candidate that minimizes the errors. In the context of regression trees, the difference between the real and the predicted values is called the residual. This loss function is very similar to the minimum squared errors (MSEs) loss function. In this case, the objective goal is to minimize these residuals.

$$J = \sum_{i=1}^{n} (real_i - pred_i)^2 \tag{1}$$

Given a set of output values in a node, each value will be tried iteratively, and the sum of squares residual will be calculated for all the threshold candidates. Once the optimized threshold is calculated, the process can be repeated by adding new nodes. When a node can no longer be split, the process stops, and it is called a leaf. Overfitting can be an issue with regression trees. To reduce the possibility of overfitting, rules are used to stop adding nodes once some criteria have been met. As an example, if a node does not have enough samples to calculate an average, perhaps if there are less than 30 samples for this node, the nodes will stop splitting.

Another key question is the selection of features to use for the decision node. Intuitively, the process is fairly logical and simple. For each feature, the optimal threshold will be calculated as previously described. Then, of the $n$ feature candidates, the one that minimizes the residuals will be selected.

### 2.2.2. AdaBoost

AdaBoost (adaptive boosting) is a more advanced version of the standard regression tree algorithm [15]. In AdaBoost, trees are created with a fixed size, which are called stumps; they can be restricted to stumps of just two levels, for instance. Some stumps have more weight than others when predicting the final value. Order is important in AdaBoost. One important characteristic in AdaBoost is that errors are considered and given more importance when creating subsequent stumps.

In AdaBoost, the training samples have weights to indicate importance. As the process advances, the weights are updated. Initially, all samples have the same weight, and all weights add up to one. During training, the weight for the stump is calculated by determining the accuracy between the predicted and the real values. The stump weight is equal to the sum of the weights of all badly (poorly) predicted samples. The weight importance is also known as the amount of say ($w_{import}$) and is calculated as follows:

$$w_{import} = \frac{1}{2} \log \frac{1 - Error}{Error},$$ (2)

where *Error* represents the total number of errors.

In the next iteration, the data sample weights need to be adjusted. Samples that have caused errors in the model have their weights increased. The other samples have their weights decreased.

The formula to increase the weights for error samples is as follows:

$$w = w \times e^{w_{import}}.$$ (3)

To decrease the weights for non-error samples, the formula is as follows:

$$w = w \times e^{-w_{import}}.$$ (4)

After updating the weights for the data samples, the weights are normalized to add up to one.

In the next iteration, a new data set is created that contains more copies of the samples that have the highest error weights. These samples can be duplicated. A special random selection algorithm is used to select these samples. Although random, the algorithm favors the selection of samples with higher error weights.

Once the new training data set is created, weights are reassigned to each sample of the new data set. The new data set is the same size as the original data set. All weights are set to be the same again, and the process begins again. The point is that penalized samples in the previous iteration are duplicated more in the new data set.

This method is how a model learns the AdaBoost stumps and their weights. Then, the model can perform predictions using all these stumps, but the learned weights are considered when making a prediction.

### 2.2.3. Gradient Boost

AdaBoost, as previously described, builds small stumps with weights. Gradient boost (as compared with AdaBoost) builds subtrees that can have one node or more [16]. However, the nodes are added iteratively. The final gradient boost tree is a linear combination of subtrees.

The first node in the linear combination of subtrees is simply one node that predicts the average of all regression outputs for the training data. The algorithm then continues to additively add subtrees. The next subtree is based on the errors from the first tree, such as the errors made in the tree that only has one node predicting the average of the *y* output values in the training data.

The errors are the differences between the predicted and the real output values. As previously described, the gradient boost tree is a linear combination of subtrees (see Equation (5)). The first node predicts the average regression output, but subsequent subtrees predict residuals. These residuals are added to the initial average node and adjust the output value. The residuals are the differences between predicted and real values. Similar to previously described regression trees, the residuals are assigned to nodes in the tree based on how they minimize errors.

$$y = subtree_0 + lr \times subtree_1 + lr \times subtree_2 + lr \times subtree_3, \tag{5}$$

where $subtree_0$ predicts the average output; all other subtrees predict residuals.

To prevent overfitting, a learning rate (*lr*) is assigned (which is a scaling factor) to the subtrees to control the effect of the residuals on the final regression output.

Every iteration, the new tree is used to predict new output values, and these values are compared with the real values. The difference is the residuals. These residuals are used to create the new subtree for the next iteration. Once the residuals for an iteration are calculated, a new subtree can be created. The process involves ranking the residuals. This process continues until a stopping criterion is reached. The learning rate is usually 0.1.

### 2.2.4. XGBoost

XGBoost [17] is an improvement on gradient boosting. It has new algorithms and approaches for creating the regression tree. It also has optimizations for improving the performance and speed by which it can learn and process data.

XGBoost has some similar characteristics to gradient boosting. The first node always predicts 0.5 instead of the average, as with gradient boosting. Also similar to gradient boosting, XGBoost fits a regression tree to calculate residuals with each subtree. Unlike gradient boosting, XGBoost has its own algorithm for building the regression tree. Formally, XGBoost [17] can be defined as follows.

For a given data set, a so-called tree ensemble model uses *K* additive functions to predict the regression output and is denoted as follows:

$$\hat{y}_i = \sum_{k=1}^{K} f_k(X_i), \tag{6}$$

where $f_k \in F$. The symbol *F* is the space of regression trees. Each $f_k$ corresponds to an independent subtree. Each regression subtree contains an output value on each leaf. This value is represented by *w*. For a given sample, the decision rules in the tree are used to find the respective leaves and calculate the final prediction. This prediction is obtained by summing up the output values in the corresponding leaves. To learn the set of functions used in the regression tree, the following loss function must be minimized.

$$J = [\sum_{i=1}^{n} L(y_i, \hat{y}_i)] + \frac{1}{2}\lambda(\|w\|)^2 + \Upsilon T, \tag{7}$$

where $L(y_i, \hat{y}_i)$ is a type of MSE measuring the difference between the real and predicted values and the term $\Upsilon T$ is used to control the number of terminal nodes. This process

is called pruning, and it is part of the optimization objective. The term $\frac{1}{2}\lambda(\|w\|)^2$ is a regularization parameter; it represents the output value $w$ and helps to smooth the learned values. Moreover, $\lambda$ is a scaling factor. Setting the regularization parameters to zero makes the loss function become the same as the traditional gradient tree-boosting loss.

The XGBoost "tree ensemble" model includes functions that are used as parameters and as such cannot be optimized using typical optimization methods. Instead, the model is trained in an additive manner. Formally [17], let $\hat{y}_i^{(t)}$ be the prediction of the instance $i$ at iteration $t$. The new $f_t$ (i.e., subtree) will need to be added to minimize the loss as follows:

$$J^{(t)} = [\sum_{i=1}^{n} L(y_i, \hat{y}_i^{t-1} + f_t(x_i))] + \frac{1}{2}\lambda(\|w\|)^2 + \Upsilon T. \qquad (8)$$

This means that an $f_t$ (subtree) must be added "greedily", which results in the most improvement to this model based on the loss. So, in XGBoost, given a node with residuals, the goal is to find output values for this node that create a subtree that minimizes the loss function. The term $\frac{1}{2}\lambda(\|w\|)^2$ can be written as $\frac{1}{2}\lambda O_{value}^2$. The term $O_{value}^2$ represents the output value.

Optimization in XGBoost is somewhat different in practice from optimization in neural networks. In neural networks, derivatives are taken for gradients during the epochs in the training process, and then the stochastic gradient descent with back propagation is performed. In contrast, in XGBoost, a loss has to be calculated and theoretically minimized; in practice, however, the derivatives (gradients) are always determined using the general equation of adding the residuals and dividing by the total. Therefore, in essence, no derivative calculation occurs during training. Unlike neural networks, XGBoost uses the same overall design for every single model, and the model only has to calculate the derivative once (on paper), where we will know that it will work for all of the models created. In contrast, every neural network can have a different design (such as having different numbers of weights, hidden layers, neurons, activations, and loss functions). Therefore, the neural network-based optimization always requires the calculation of the gradient for each model during each training epoch. Thus, in general, by minimizing the loss function with the derivative once, general XGBoost equations are obtained, which are used to build the tree. This derivation supplies the general formulas. This objective can be optimized in a general setting. Derivation and simplification provide general equations.

In normal circumstances, it is not possible to enumerate all possible tree structures. Instead, a greedy algorithm that starts from a single leaf and iteratively adds branches to the tree is used. As such, the process to create the tree is as follows.

In essence, different subtree candidates must be tried, and the one that optimizes the loss function should be selected. During the learning process, the tree starts with one node and tries to add more subtrees with optimal node and threshold configurations. The approach needs a way to score the subtree candidates to select the best one. Each subtree is scored using the "gain" value. The "gain" value for a subtree candidate is calculated from other values that are associated with each of the nodes, which make up that subtree candidate. Once the "similarity" scores for each node in the subtree candidate are obtained, they are used to calculate the "gain" of that particular split of the nodes in the subtree. This "gain" equation is used to evaluate the split candidates. For example, assuming that there are nodes left and right for a root node, the "gain" can be calculated as follows:

$$gain = left\_sim\_score + right\_sim\_score - root\_sim\_score. \qquad (9)$$

The "gain" score helps to determine the correct threshold. For other threshold cutoffs, other gains can be calculated and the optimal one selected. Thus, the threshold that provides the largest "gain" is used. If a node only has one residual, then that is all, and it is a leaf.

The "gain" depends on "similarity" scores. The process to create a set of subtree candidates needs these "similarity" scores and is conducted as follows. For each node in a

subtree candidate, a quality score or "similarity" score is calculated from all residuals in the node. The "similarity" score function is as follows:

$$sim\_score = \frac{(sum\_of\_residuals)^2}{number\_of\_residuals + \lambda},$$ (10)

where the residuals are first summed and then squared. The symbol $\lambda$ is a regularization parameter.

Once the "similarity" score is calculated, the node should be split into two child nodes. The residuals are grouped into these nodes. Because the threshold for the node is not known, the feature is iteratively split from its lowest value to its highest values in the range. Here, however, XGBoost uses an optimized technique called *percent quantiles* to select the threshold. Residuals that belong to less than the threshold go on the left node, and residuals that belong to more than the threshold go on the right node. Once again, the similarity scores are calculated for both the left and right nodes.

The optimal subtree candidate is selected using the "similarity" score and the "gain" score as previously described. Once the best candidate is selected, the next step is to calculate its output values for the leaf nodes. The output values are also governed by the loss function, and the formula for them can be derived from it, as shown in Equation (11). So, for a given subtree structure, the optimal $w_j$ of leaf $j$ can be computed as

$$O_{value} = \frac{sum\_of\_residuals}{number\_of\_residuals + \lambda}.$$ (11)

This step completes the tree-building process. Now, the tree can be used to predict $y$ similarly to how it is performed with gradient boosting.

$$y = subtree_0 + lr \times subtree_1 + lr \times subtree_2 + lr \times subtree_3,$$ (12)

where $subtree_0$ predicts 0.5 and all other subtrees predict residuals that are added together but are scaled based on the learning rate $lr$.

### 2.3. Neural Networks for Regression

Neural networks can be used for regression modeling. Neural networks are universal function approximators. The function consists of an input, an output, and one or more hidden layers. They can approximate nonlinear data using activation functions such as the sigmoid function. The weights in the neural network are learned through stochastic gradient descent and back propagation. The goal is to optimize a loss function. For regression, the most common loss function is MSE.

## 3. Methodology

In this section, the data set, modeling methodologies, and performance metrics are presented and discussed in addition to an overview of the CFD modeling techniques used to simulate the blast furnace process.

### 3.1. Computational Fluid Dynamics Modeling of the Blast Furnace

CFD modeling techniques provide unique insight into the interior state of the blast furnace, with a validated model being capable of acting as a suite of soft sensors. The earliest applications of CFD to the blast furnace employed significant simplifications for physics and geometry [18–20]; however, advancements in computing capacity and technology have enabled increasingly complex modeling of combustion, heat transfer, turbulent and multiphase flow, and the other physical phenomena occurring within the furnace using real-world geometry [21–24]. Previous research conducted by researchers at Purdue University Northwest with validated in-house CFD solvers tailor-designed for blast furnace modeling have explored a wide range of conditions and their effects on operation, including developing guidance for fuel injection, lance design and positioning, overheat troubleshooting,

and coke consumption rate reduction [25]. The modeling approach employed to generate data for this work simulates conditions within the blast furnace blowpipe and tuyere (hot blast flow and fuel injection into the furnace), raceway (fuel combustion and reducing gas generation), and shaft (chemical reduction reactions and iron melting). Full details for these models can be found in previous publications [25–27]. This paper will describe the core elements only in brief. Fluid flow, heat and mass transfer, and chemical reactions in all zones are simulated using the standard Navier–Stokes equations, the k-$\epsilon$ turbulence model, and the species transport equation, discretized with the Semi-Implicit Method for Pressure-Linked Equations scheme. The general form of the governing PDEs used are in Table 1.

**Table 1.** CFD blast furnace model governing equations.

| | |
|---|---|
| Cons. of mass: $\phi = 1$ | |
| Cons. of momentum: $\phi = velocity$ | $\nabla \cdot (\rho \phi u) = \nabla \cdot (\Gamma_\phi \nabla \phi) - \nabla \cdot (\rho u^t \phi^t) + S_\phi$ |
| Cons. of energy: $\phi = enthalpy$ | |
| Species transport | $\nabla \cdot (\rho u Y_i) = \nabla \cdot (\rho \Gamma_i \nabla Y_i) + R_i + S_i$ |
| Turbulent kinetic energy | $\nabla \cdot (\rho k u^t) = \nabla \cdot (\frac{\mu_t}{\sigma_k} \nabla k) + 2\mu_t S_{ij}.S_{ij} - \rho\varepsilon$ |
| Turbulence dissipation rate | $\nabla \cdot (\rho \varepsilon u^t) = \nabla \cdot (\frac{\mu_t}{\sigma_\varepsilon} \nabla \varepsilon) + C_{1\varepsilon} \frac{\varepsilon}{k} 2\mu_t S_{ij}.S_{ij} - C_{2\varepsilon}\rho\frac{\varepsilon^2}{k}$ |
| Where: | $\mu_t = C\rho vl = \rho C_\mu \frac{k^2}{\varepsilon}, C_\mu = 0.09, \sigma_k = 1.00, \sigma_\varepsilon = 1.30,$ $C_{1\varepsilon} = 1.44,$ and $C_{2\varepsilon} = 1.92$ |

Here, $\rho$ is density, $u$ is velocity, $\phi$ is the general property transported, $u^t$ and $\phi^t$ represent the fluctuating components of velocity and the transported property owing to turbulence, and $S_\phi$ is a source term. $Y_i$ is the local mass fraction of species, $i$; $R_i$ is the net production rate of that species; $\Gamma_i$ is the species diffusion coefficient; and $S_i$ is a source term for species creation. $k$ and $\epsilon$ are the turbulent kinetic energy and dissipation rate, respectively, $\mu_t$ is the turbulent viscosity, and $C_\mu$, $\sigma_k$, $\sigma_\epsilon$, $C_{1\epsilon}$, and $C_{2\epsilon}$ are model constants that are defined as listed in Table 1. In total, 6 key chemical reactions are predicted in the tuyere and raceway regions and 11 are predicted in the shaft, including gas combustion, the Boudouard reaction, indirect reduction of iron oxides by CO and $H_2$, and more.

CFD Model Validation and Case Matrix Scenarios

It should again be noted that although the capabilities of applied CFD modeling have been illuminated by the aforementioned studies and by similar studies, even modern multicore PCs and high-performance computing clusters are not yet powerful enough to run simulations in near-real time. It remains a challenge to provide this high-fidelity, simulation-based guidance to operators quickly enough for day-to-day operations. As this effort aims to develop a reduced-order model for a single blast furnace that is capable of providing equivalent operational predictions to CFD modeling in near-real time, a CFD model of the selected site blast furnace was developed and validated against real-world operating conditions that are typical for North American blast furnaces. The model was then used to predict furnace performance at a preselected range of operating conditions that are relevant to industrial practice.

Validation was conducted against averages from a 2-week period of real-world operating conditions at a North American blast furnace. CFD results were compared against available data, including pressure, fuel and gas flow rates, gas temperatures, top gas analyzers, material charge records, and a daily production and coke rate estimation. Some assumptions were made in the modeling to account for information not available or not

recorded in real-world operation, including the burden bulk porosity distribution and moisture content of charged material. Table 2 shows the comparison between simulation predictions for the provided operating conditions and the monthly averages of the 2-week operation period.

**Table 2.** CFD model validation for the North American blast furnace.

|  | $\Delta P$ (kPa) | Top Gas Temp. (K) | Coke Rate (lb/ton of HM) | CO Utilization |
|---|---|---|---|---|
| CFD | 109 | 391 | 925 | 47.2 |
| Industrial data | $\sim$115 | $\sim$370 | $\sim$926 | 46.8 |
| Difference (%) | 5.6 | 5.9 | 0.06 | 0.85 |

The variables that were selected to generate the matrix of scenarios for training reduced-order models were the natural gas injection rate, natural gas injection temperature, pulverized coal injection rate, $H_2$ injection rate, charged ore moisture content, and oxygen enrichment. The input variables have been normalized so as not to publish the specific operating conditions at the industrial furnace. All other variables at the tuyere level (including factors such as wind rate, steam injection, and ambient moisture) are held constant; however, modifications to the charged ore/coke ratio are made to account for the corresponding changes, which would occur in real-world operation with different fuel injection rates.

### 3.2. Regression with Neural Networks

The architecture for the neural network-based models used in our work consists of a regression function, which is the sum of a linear function and a nonlinear function $f(x)$, as described in Equation (13). The nonlinear part is a simple neural net (NN) with one hidden layer consisting of 10 neurons. The hidden layer uses a sigmoid activation function. Both the inputs and outputs were scaled using standardization.

$$g(x) = w * x + b + f(x). \tag{13}$$

The loss function is the standard MSE.

### 3.3. Regression with Neural Networks and Probability Density Function Shaping

Probability density function (PDF) shaping [28,29] is a technique that can help a neural network-based regression model learn while taking into consideration the error distribution in the predicted data. The goal is to make sure that the histogram of the errors in the predicted data has a Gaussian distribution that is as close to a zero mean as possible and with as narrow a standard deviation as possible. In industrial processes, it is a good reliability and quality safeguard.

### 3.4. Regression with XGBoost

XGBoost is a tree-based gradient boosting technique that was described in Section 2. It has been shown to achieve good results on tabular and regression problems that have small amounts of data. XGBoost has been extensively used in Kaggle competitions with excellent performance; however, it does have well-known problems when dealing with cases that require extrapolation from the trained data ranges.

### 3.5. Data and Features

The data set [30] consisted of 894 samples, which were split into training and testing sets. An 80% split was used after randomizing the data, which resulted in 715 samples for training and 179 for testing. There were eight key inputs and, at most, six outputs per each of the three reaction regions of the blast furnace as represented by the CFD models (tuyere, raceway, and shaft). The CFD modeling techniques use input features that are directly representative of the operating conditions for the blast furnace. The list of input

variables are shown at the upper section of Table 3. The inputs are normalized; however, the typical ranges for blast furnace operating conditions can be found in the published reference materials [31,32]. North American blast furnaces of similar scale and production range between 2000 and 3500 cu. meters in working volume, with hearth diameters of 10+ meters, and 30+ tuyeres. As an example of the operating conditions that are typically used in North American blast furnaces, blast air oxygen enrichment ranges from 23 to 32%, natural gas and pulverized coal are injected at rates ranging from 30 to 130 kg per ton of hot metal produced, and hot blast temperatures range from 1300 K to 1450 K. The CFD approach can generate a wide range of potential output data, which are predicted based on chemical reactions, mass and energy transfer, and the fluid and solid flow within the process. Those selected for this study were identified as key output variables that would influence the selection of operating conditions for stable production at an industrial blast furnace. The outputs included in this study are listed in the lower section of Table 3. Moreover, the full data set is available from the project repo [30].

**Table 3.** Input and output variables of the CFD model used in this study.

| Category | Variable | Definition |
|---|---|---|
| Input | i_h2_inj_kg_thm | $H_2$ injection rate in kg per ton of hot metal |
| Input | i_pul_coal_inj_kg_thm | Pulverized coal injection rate in kg per ton of hot metal |
| Input | i_nat_gas_inj_kg_thm | Natural gas injection rate in kg per ton of hot metal |
| Input | i_nat_gas_t_k | Natural gas injection temperature in Kelvin |
| Input | i_o2_vol_perce | Blast oxygen enrichment in % |
| Input | i_hot_blast_temp_k | Hot blast temperature in Kelvin |
| Input | i_ore_moisture_weight_perce | Ore moisture content by weight in % |
| Input | i_ore_weight_kg | Weight of iron ore charged per layer in kg |
| Output | o_tuyere_exit_velo_m_s | Tuyere exit velocity in meters per second |
| Output | o_raceway_flame_temp_k | Raceway flame temperature in Kelvin |
| Output | o_raceway_coal_burn_perce | Pulverized coal burnout in % |
| Output | o_raceway_volume_m | Raceway volume in cubic meters |
| Output | o_shaft_co_utiliz | CO use in % |
| Output | o_shaft_$H_2$_utiliz | $H_2$ use in % |
| Output | o_shaft_top_gas_temp_c | Blast furnace top gas temperature in Celcius |
| Output | o_shaft_press_drop_pa | Shaft region pressure drop in Pascals |
| Output | o_shaft_coke_rate_kg_thm | Change in BF coke consumption in kg per ton of hot metal |
| Output | o_shaft_gasspecies_v_perc | Top gas CO, $CO_2$, $H_2$, $N_2$ volume % |

*3.6. Performance Metrics*

The $R^2$ metric is a measure of performance that is used for regression models. Formally, it is called the coefficient of determination. It can be described [33] as a metric representing the proportion of the variation in the dependent variable that is caused or affected by the independent variable(s). For this project, the closer the $R^2$ is to 1.0, the better.

**4. Results**

In this section, the results are presented and discussed. In general, the performance was good and the models learned quickly, as shown in Figure 1.

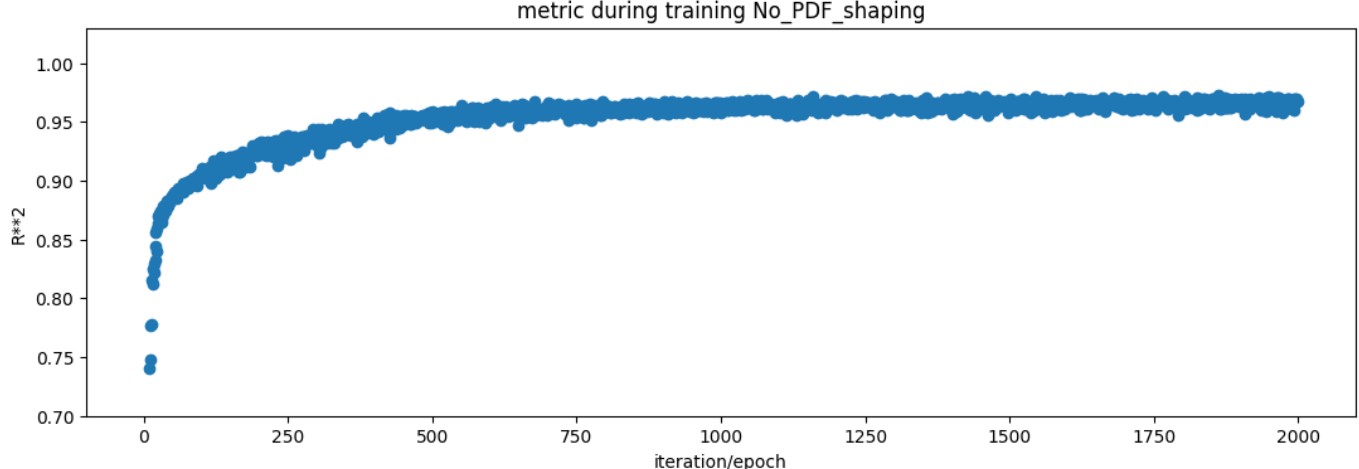

**Figure 1.** Performance tuyere $t_k$.

### 4.1. Regression Models Comparison

This section presents a comparison of the performance across all tested models. As shown in the results, both models performed exceptionally. XGBoost and NNs in some cases had the same performance. In cases where the neural nets were not able to learn, the XGBoost algorithm performed the best. Table 4 shows the results of multioutput models with NNs as well as single-output models with both NNs and XGBoost. In general, it is observed that the single-output models performed slightly better than multioutput models in like-for-like comparisons. Additionally, the XGBoost algorithm showed the best performance overall (within the range of data simulated with the CFD models).

**Table 4.** $R^2$ performance.

| Output | Multi-Output NN | Multi-Output NN-PDF | Single-Output NN | Single-Output NN-PDF | Single-Output XGBoost |
|---|---|---|---|---|---|
| Tuyere exit velocity | 0.97 | 0.96 | 0.98 | 0.96 | 0.99 |
| Tuyere exit temp. | 0.95 | 0.96 | 0.95 | 0.93 | 0.99 |
| Raceway flame temp. | 0.98 | 0.98 | 0.99 | 0.96 | 0.99 |
| Raceway coal burn | 0.83 | 0.68 | 0.89 | 0.86 | 0.99 |
| Raceway volume | 0.77 | 0.73 | 0.94 | 0.91 | 0.99 |
| $H_2$ use | 0.83 | 0.82 | 0.80 | 0.78 | 0.85 |
| Top gas temp. | 0.95 | 0.95 | 0.98 | 0.98 | 0.98 |
| Shaft pressure drop | 0.88 | 0.88 | 0.86 | 0.68 | 0.92 |
| Shaft coke rate | 0.98 | 0.98 | 0.98 | 0.96 | 0.99 |
| Top gas CO vol % | 0.90 | 0.87 | 0.92 | 0.81 | 0.92 |
| Top gas $CO_2$ vol % | 0.93 | 0.93 | 0.95 | 0.91 | 0.97 |

### 4.2. Regression Error Mappings of Predicted vs. Real Test Set

In this section, the plots of the mapping between real values vs. model-predicted values are presented and discussed. Approximately 170 test samples were run by the model and compared with the real CFD-generated test samples. The results are shown in Figure 2.

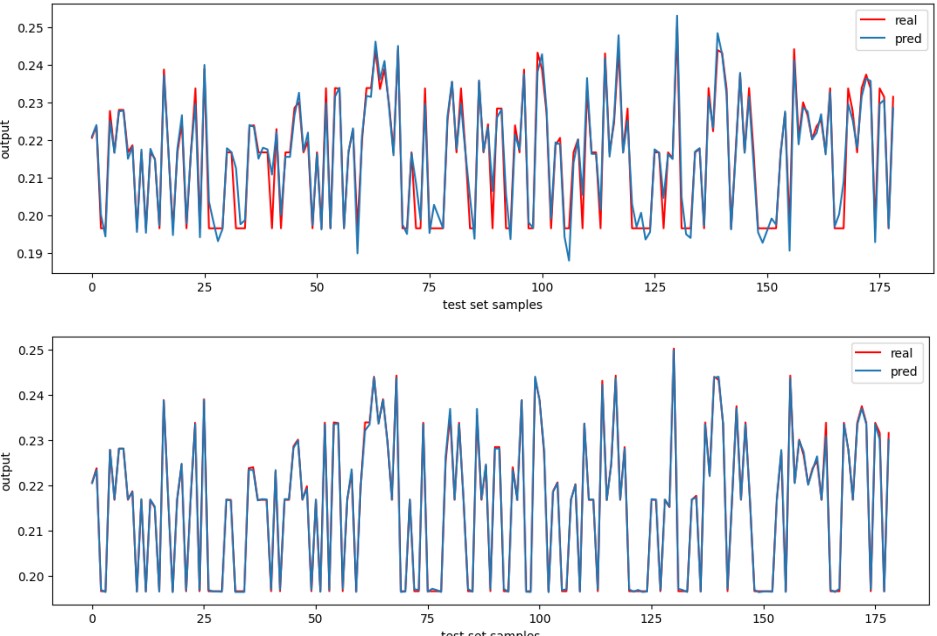

**Figure 2.** Raceway volume error mapping: (**upper**) NN; (**lower**) XGBoost.

The figure for the raceway volume output shows that the XGBoost model fits the data better than the NN model. The figure includes two plots. The top plot shows the mapping between the predicted and real values when using the neural network-based model. The bottom plot shows the mapping between predicted and real values when using the XGBoost-based model. For this case, XGBoost seems to provide better results.

Figure 3 shows the comparison for the raceway coal burn percent. The top plot shows the mapping between the predicted and real values when using the neural network-based model. The bottom plot shows the mapping between the predicted and real values when using the XGBoost-based model. Again, XGBoost fits the data better.

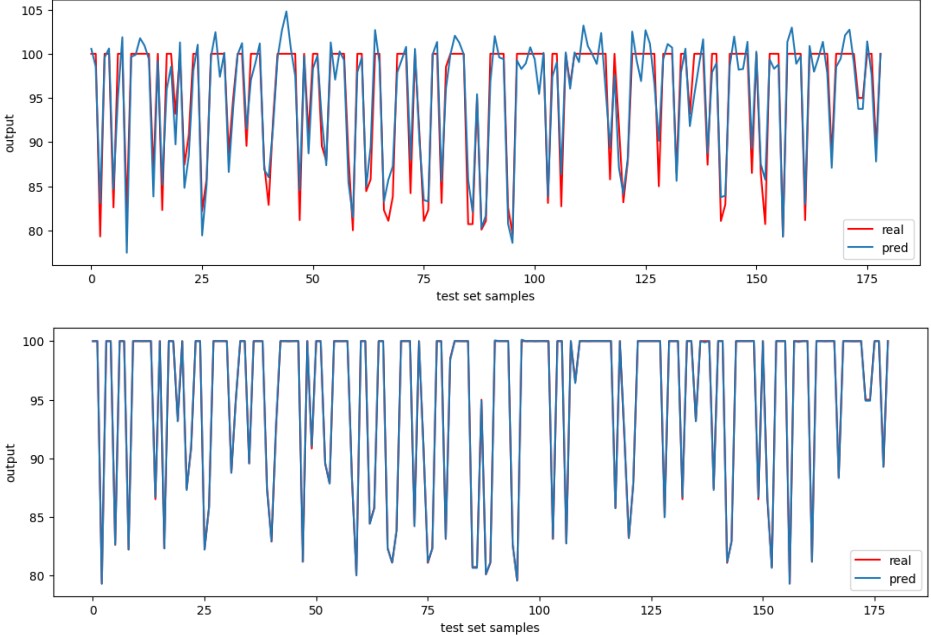

**Figure 3.** Raceway coal burn percent mapping: (**upper**) NN; (**lower**) XGBoost.

Finally, Figure 4 shows the comparison for the raceway flame temperature in Kelvin between XGBoost and the NNs. In this case, it is a bit more difficult to notice the difference.

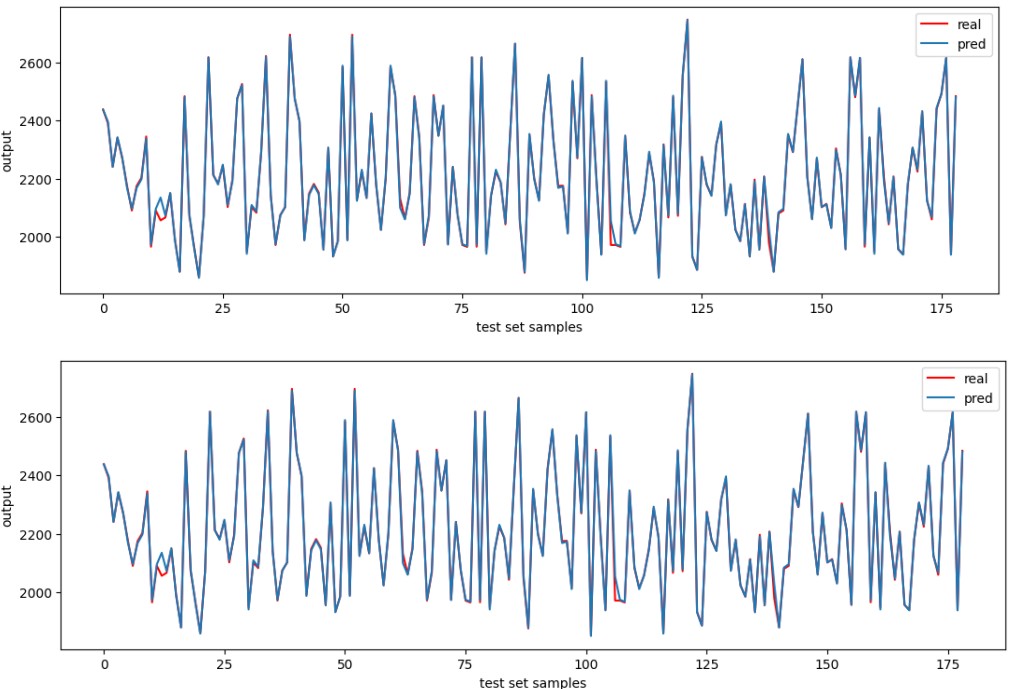

**Figure 4.** Raceway flame temperature in Kelvin mapping: (**upper**) NN; (**lower**) XGBoost.

*4.3. Regression Error Distribution Analysis for the Test Set*

In this section, an analysis of the error distributions is performed. This section shows the comparison of error distribution for the tuyere exit velocity in meters per second. A direct comparison between the two models used is shown in Figure 5 below. In general, XGBoost seems to fit a model with a predicted error PDF that has a mean of zero and a very narrow standard deviation. In contrast, the neural network model seems to have a broader standard deviation.

Figure 6 shows the comparison for tuyere *tk*, which is similar to the previous case. Although the standard deviation can vary, in general, the errors always had a PDF with a mean of zero.

*4.4. Correlations Matrix Analysis*

It is a relatively simple matter to extract correlation coefficients between the parameters included in the physics-based modeling data. This section presents the correlation coefficient matrices as a means of analyzing the relationship between input and output variables that are relevant to blast furnace operation. Correlation coefficient matrices are independent of the model used and help to identify which inputs have the highest contribution to the quality of the predictive model. Additionally, for the specific scenario of examining blast furnace operation, these correlations can be compared with existing rules of thumb that are established by blast furnace operators over decades of operation to build confidence in the validity of the models [31].

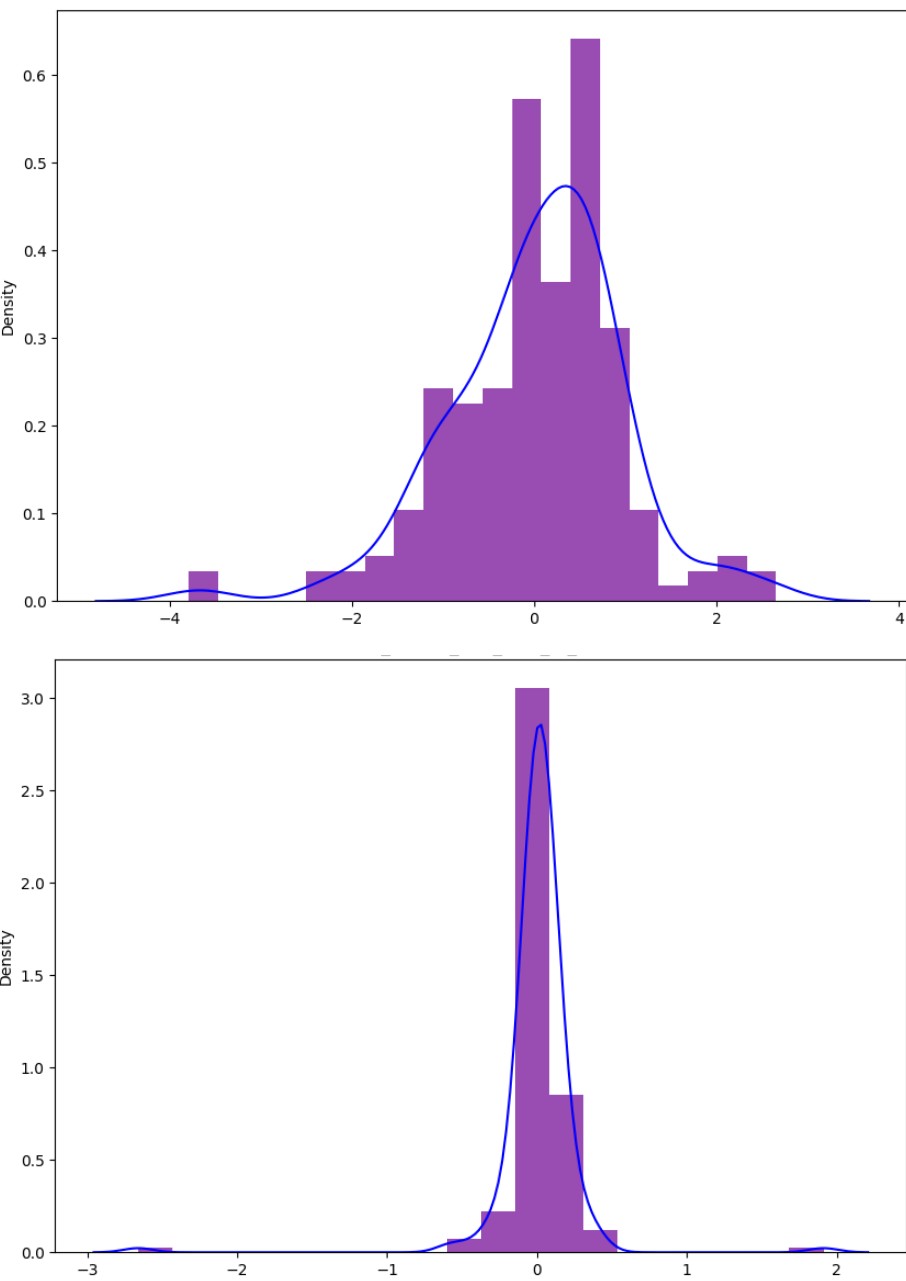

**Figure 5.** Tuyere exit velocity in meters per second: (**Upper**) NN; (**Lower**) XGBoost.

The correlation coefficients for tuyere exit velocity are explored in Figure 7. Of the parameters considered in this study, increased tuyere exit velocity is most strongly correlated with increased natural gas injection rates, increased natural gas injection temperature, and increased $H_2$ injection rates. Increased gas injection rates (natural gas and hydrogen gas) result in a higher gas volume and higher tuyere exit velocity. Increased natural gas injection temperature, supplied by preheating the injected gas, would also result in a lower-density, higher-volume gas, necessitating increased velocity. It is observed that increasing pulverized coal injection in the tuyere results in a decline in tuyere exit velocity. This result is because the blast air must accelerate the comparably much slower stream of coal particles, losing momentum in the process.

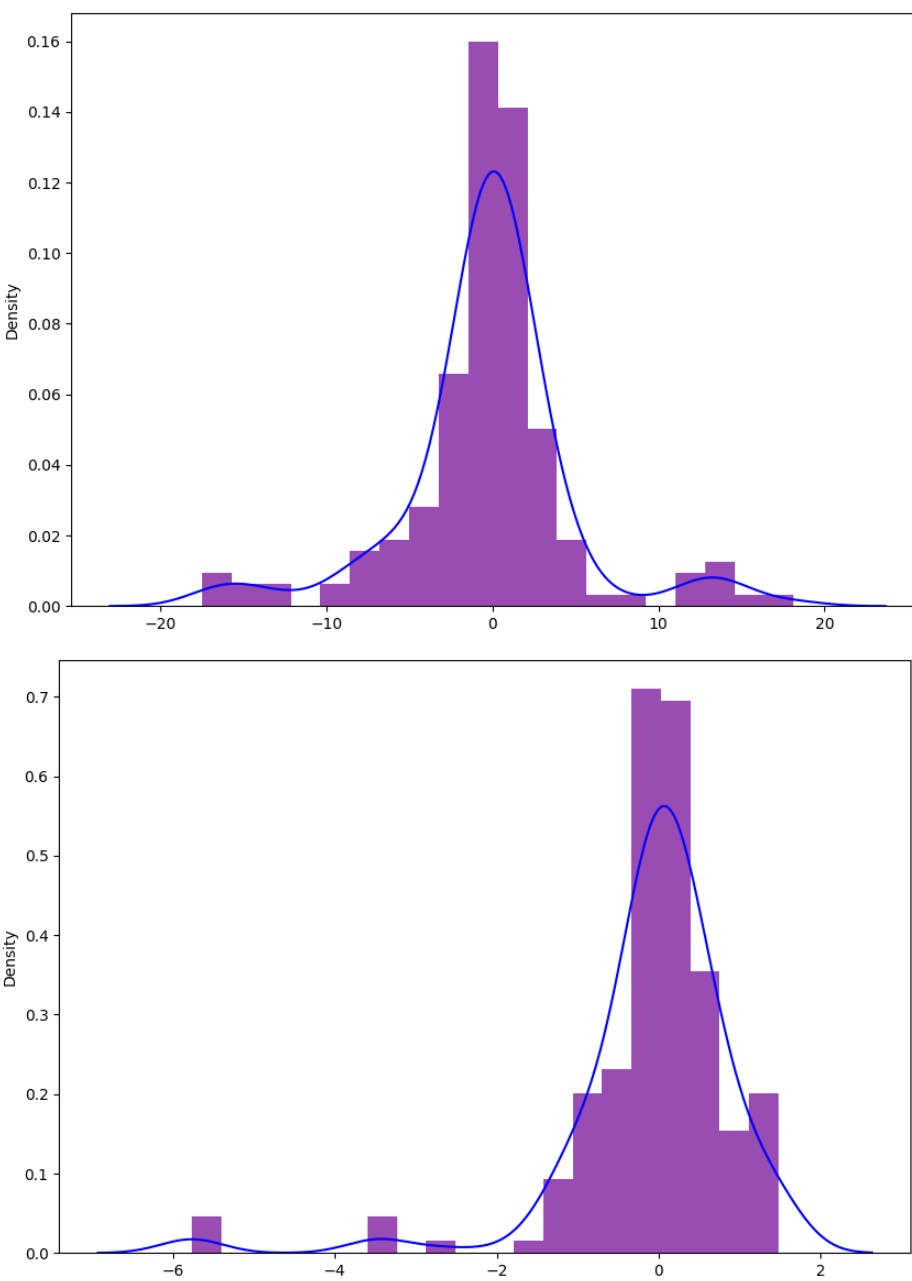

**Figure 6.** (**Upper**) NN; (**Lower**) XGBoost.

Similarly, the correlation coefficients for the CFD-predicted blast furnace raceway flame temperature are detailed in Figure 8. The key parameters positively influencing the temperature of gas generated by combustion in this lower region of the furnace are pulverized coal injection, oxygen enrichment, and hot blast temperature. These parameters make physical sense, as increasing the available carbon fuel, oxygen, and sensible heat results in higher total gas temperatures after the reactions are complete. The natural gas injection rate is inversely (strongly) correlated with this flame temperature value. This result is primarily because natural gas combusts to produce $CO_2$ and $H_2O$ gas, both of which participate in endothermic reactions with carbon coke, consuming heat to produce $CO$ and $H_2$. This reaction results in lower total gas temperatures when injecting high levels of natural gas. Of the variables highlighted here, raceway flame temperature is perhaps the most significant, as it directly influences the rate of material melting and chemical reduction reactions occurring within the blast furnace.

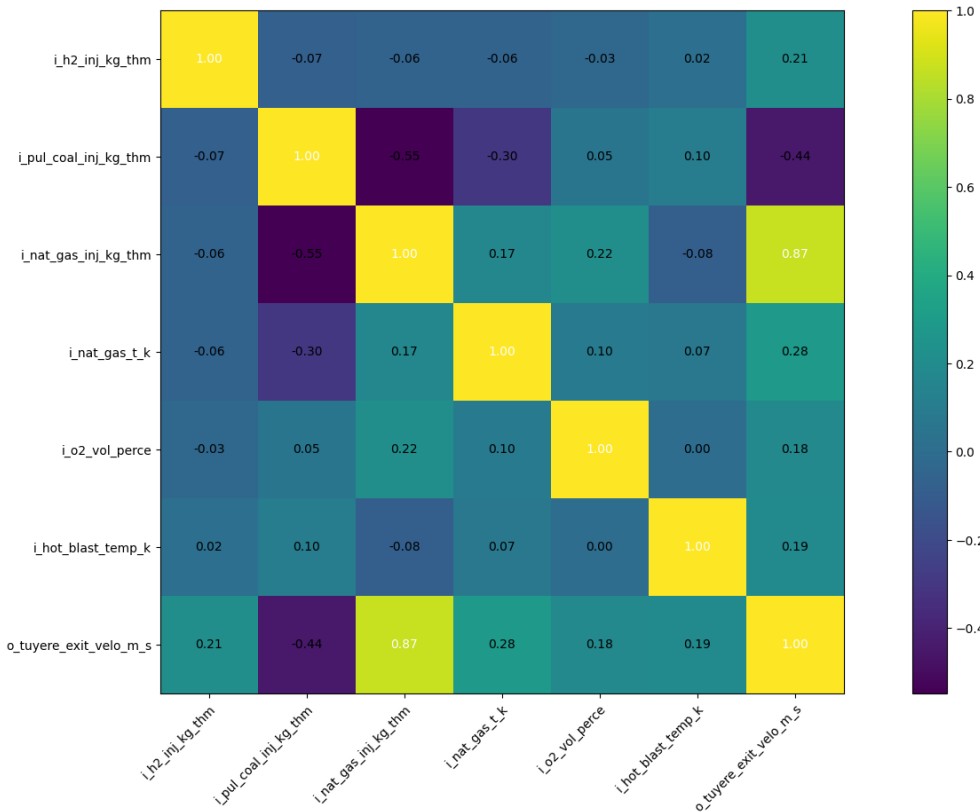

**Figure 7.** Correlation coefficient matrix for tuyere exit velocity (m/s). Input and output variables are described in Table 3.

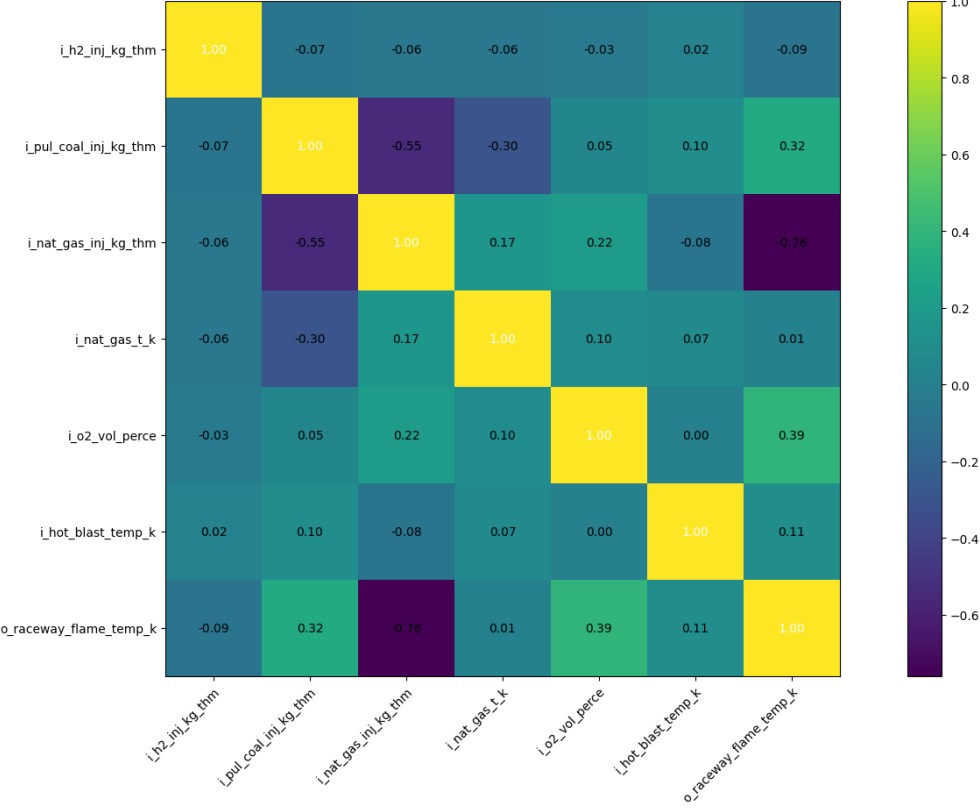

**Figure 8.** Correlation coefficient matrix for raceway flame temperature. Input and output variables are described in Table 3.

As a final example, the correlation coefficients for pulverized coal burnout (Figure 9) indicate that coal burnout is most strongly correlated (inversely) with the coal injection rate. Because the rate of coal combustion is most strongly limited by the heating of the coal, the delivery of pulverized coal at high rates can results in low overall burnout. Not enough oxygen is available to combust the coal outside the raceway, so quickly heating the particles to combustion temperatures before they affect the coke bed is crucial to combustion efficiency. Increases in the natural gas injection rate, natural gas temperature, and oxygen enrichment result in increased pulverized coal burnout for the same reason, as each of these variables increases the rate at which pulverized coal is heated and engages combustion more rapidly.

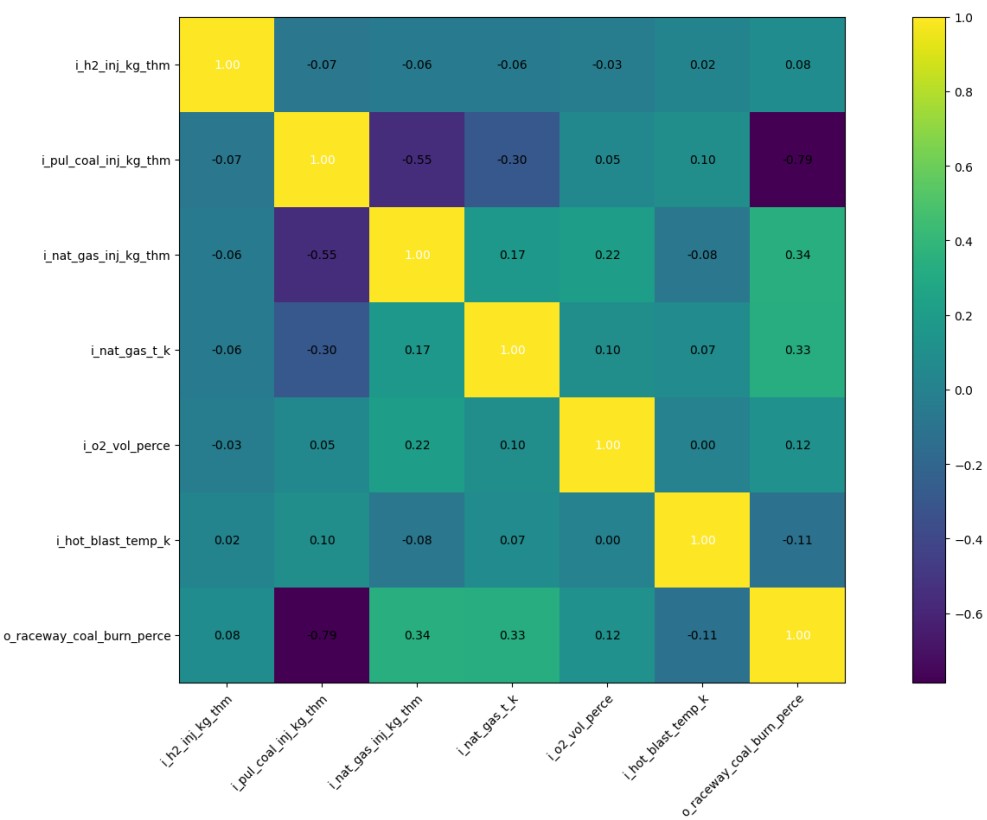

**Figure 9.** Correlation coefficient matrix for pulverized coal burnout. Input and output variables are described in Table 3.

## 5. Discussion

Notably, XGBoost is effective for this type of problem; however, XGBoost models do not perform well for extrapolation applications. In general, the neural network-based models provide more realistic predictions for data outside of the training ranges and as such provide the more robust option for operators to utilize between the compared models discussed here. Additionally, using both models together (XGBoost within the simulated range and neural network based models for extrapolation) makes sense for this use case and probably for other industrial applications where both precision and the ability to extrapolate and predict for never-before-seen cases are desired.

## 6. Conclusions

In this work, the authors have used machine learning-based techniques to model certain sections of an ironmaking blast furnace. In particular, regression models using neural networks and XGBoost have been trained and tested. Analysis based on $R^2$ statistics shows that the models perform well in the prediction of physics-based data that are generated by CFD modeling of the blast furnace process. Comparison with validated

CFD modeling, operational data from the blast furnace in question, and additional subject matter expert analysis have all confirmed that the results of the models appear accurate to real-world operation. It was noticed that although XGBoost sometimes achieved better results, the model showed poorer effectiveness at extrapolation. In this case, even if less accurate, a neural network model is preferred to ensure a base level of realism in the predictions beyond the data sample range.

Future work will include adding more data samples as well as generating more specific case scenarios. Finally, the effect of including real-world blast furnace sensor data in addition to simulation modeling results in the training data set will also be explored.

**Author Contributions:** Conceptualization, H.W. and T.O.; methodology, R.A.C. and H.W.; software, R.A.C.; validation, T.O. and O.U.; formal analysis, all authors; data creation, O.U.; writing—original draft preparation, R.A.C.; writing—review and editing, R.A.C. and T.O.; visualization, R.A.C.; supervision, T.O.; project administration, T.O.; funding acquisition, T.O. All authors have read and agreed to the published version of the manuscript.

**Funding:** This research was funded by DOE grant number DE-EE0009390.

**Institutional Review Board Statement:** Not applicable.

**Informed Consent Statement:** Not applicable.

**Data Availability Statement:** The data used in this work are available on GitHub at https://github.com/rcalix1/ProbabilityDensityFunctionsFromNeuralNets accessed on (23 September 2023).

**Acknowledgments:** This research was supported by the US Department of Energy's Office of Energy Efficiency and Renewable Energy under the Industrial Efficiency and Decarbonization Office Award Number DE-EE0009390. The authors would like to thank the members of the Steel Manufacturing Simulation and Visualization Consortium for their support on this effort. Support from the staff and students at Purdue University Northwest and the Center for Innovation through Visualization and Simulation is also appreciated.

**Conflicts of Interest:** The authors declare no conflict of interest.

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
