# Peer review of "Machine Learning-Based Regression Models for Ironmaking Blast Furnace Automation"

_2673-8716, doi:10.3390/dynamics3040034_

Round 1

Reviewer 1 Report

The authors have used machine learning–based techniques to model operation parameters in certain sections of an ironmaking blast furnace.

The topic is very important to metallurgists and ironmaking research community. The topic fits the scope of the journal Dynamics.

The results of this manuscript are well presented and organized. The presented results of the performed modeling work have scientific meaning.

I recommend to accept this manuscript after the sufficient revisions.

1.       In many part, the terminology ironmaking blast furnace was used. Why not revise the tittle of this manuscript Steel Furnace to Ironmaking Blast Furnace?

2.       In introduction part, the authors may address more on the significance of parameters. Which parameter is most important and the significance of those parameters may be addressed in introduction part. I learned from manuscript, the tuyere exit velocity, raceway flame temperature, and pulverized coal burnout are focus. What are the functions of those parameters in ironmaking process?

3.       The authors described a lot on CFD. What about machine learning study on the data from industries? If this is not possible. A description or referred validation of the CFD models should be given. As the reviewer understood, the accuracy of model predictions are based on the number of sets of data and the fluctuation of data. The CFD data follows a rule from CFD software simulation, i.e. finite volume or chemical reaction equilibrium and they are somehow easy to model by machine learning method. Industrial data are more changeling.

4.       Some references on the machine learning of industrial blast furnace should be referred and reviewed. For example, the references of Henrik Saxen.

5.       For a better application perspective, please given modeling blast furnace details for example but not limited to volume of blast furnace, the shape of shaft, top, and bottom of BF (graphically), arrangement and number of tuyeres.

6.       Again, a common or typical results for example pulverized coal, natural gas, H2 injection rate, and the temperature of hot blast air should be given. In other words, some of the typical data of input parameters.

7.       Noticed the input parameters in table 3 are mostly operational parameters. The output parameters are results or say output of process parameters. That’s reasonable.

8.       The tuyere exit velocity were compared in the manuscript as an important parameter. I thought this is not a big issue if the data were generated from CFD models.

Author Response

Please see the uploaded file on our point-by-point response - thanks .

Reviewer 2 Report

Summary:

This study explores the application of machine learning techniques to model sections of a steel blast furnace, traditionally simulated using Computational Fluid Dynamics (CFD). By training regression models with data from CFD simulations, the authors compare neural network-based models with XGBoost. These models exhibit strong predictive accuracy, promising improvements in real-time decision-making for blast furnace operation.

Comments:

-The manuscript appears to have several references that are missing, which should be reviewed, and citations added where applicable. In the introduction section, it's noticeable that only two references ([1] and [2]) have been utilized, and it would greatly benefit from a more comprehensive literature review. Additionally, there is a citation order discrepancy, with citations [7–9] mentioned on line 282, while references [3] and [4] are cited later on line 346. It's essential to ensure consistency and accuracy in referencing throughout the paper.

-To avoid plagiarism, it's crucial to provide proper references or any information drawn from existing literature.

-The introduction of the paper indeed appears to be relatively short and lacks a comprehensive literature review. To enhance the introduction, it is necessary to add relevant studies and their findings from the existing literature. Therefore, more studies needed to be reviewed and added to the introduction by mentioning their main studied parameter and their key findings.

-The title of Section Two, "2. Literature Review," should be revised since it primarily focuses on elucidating the models rather than conducting a conventional literature review of papers.

- What factors or features in the CFD data were found to be particularly influential in training the machine learning models?"

Author Response

Please see the uploaded file on our point-by-point response - thanks,

Round 2

Reviewer 1 Report

The modeling parameters are from reference 21-25, however, those results are not identical. The authors should refer a specific BF as supposed in a US BF. Reference 32 is a general notebook on blast furnace operation. Some typical input value or at least tuyere numbers and blast furnace. Volume should be given. Those parameters are very common and could be given. Otherwise how could we believe your results?

Author Response

Thanks and we have further modified the paper with the following details which have been highlighted in the pdf document of the revised paper.

Section 3.5 in the manuscript has now been edited to include some general operating conditions and dimensions for North American blast furnaces, in addition to a reference to AIST's BF roundup document from 2022. Since a full dataset on predicted furnace performance is published alongside the paper, the authors adhere to a request from the industrial collaborator not to disclose the specific blast furnace modeled or to directly disclose the specific operating conditions corresponding to furnace performance levels in the CFD-generated dataset in the interests of operational confidentiality. A general range has been provided for tuyere number, working volume, and some key operating conditions.

Reviewer 2 Report

Thank you for your revision. The manuscript can be accepted in its current form.

Author Response

Thanks for accepting our modifications and your time to review the paper.